# Effect of Rehabilitation Nutrition on a Post-Acute Severe COVID-19 Patient: A Case Report

**DOI:** 10.3390/healthcare9081034

**Published:** 2021-08-12

**Authors:** Kengo Shirado, Yuji Furuno, Kozue Kanamaru, Naoto Kawabata, Shota Okuno, Toshihiro Yamashita

**Affiliations:** 1Department of Rehabilitation, Aso Iizuka Hospital, 3-83 Yoshiomachi, Iizuka, Fukuoka 820-8505, Japan; yfurunoh2@aih-net.com (Y.F.); nkawabatah1@aih-net.com (N.K.); sokunoh1@aih-net.com (S.O.); tyamashitah8@aih-net.com (T.Y.); 2Department of Nutrition, Aso Iizuka Hospital, 3-83 Yoshiomachi, Iizuka, Fukuoka 820-8505, Japan; kkanamaruh1@aih-net.com

**Keywords:** case reports, exercise, nutrition therapy, sarcopenia, SARS-CoV-2

## Abstract

Coronavirus disease 2019 (COVID-19) may lead to post-acute physical function deterioration due to intensive-care-unit-acquired weakness-related sarcopenia and dyspnea. Limited reports have examined the effects of rehabilitation and nutritional therapy on patients with post-acute COVID-19. We present the case of a 67-year-old man, who was admitted for the treatment of post-acute severe COVID-19, who benefited from rehabilitation nutrition. When the patient’s condition stabilized, sarcopenia and malnutrition were observed, and rehabilitation nutrition was implemented. The physical therapist implemented a program focused mainly on resistance training and aerobic exercise, and the dietitian provided oral nutritional supplements and hospital food that met the patient’s energy and protein intake requirements. Comparing the initial evaluations with those at discharge, factors affecting nutritional status, such as body mass index and skeletal muscle mass index, and physical functions, such as grip strength and walking speed, and dyspnea, had improved. The patient was discharged and returned to work. This case suggests improvements in the nutritional status and physical functions of post-acute severe COVID-19 patients by interventions following rehabilitation nutrition.

## 1. Introduction

Coronavirus disease 2019 (COVID-19) was first reported in December 2019 and has since become a global pandemic. Some patients with severe COVID-19 are at a high risk of developing intensive-care-unit-acquired weakness (ICU-AW) [1], and even after 1 month of recovery from fever, some patients still develop dyspnea [2] and muscle weakness [3]. Post-symptom resolution, sarcopenia may develop from sequelae specific to COVID-19 patients and/or ICU-AW.

The effects of rehabilitation on post-acute COVID-19 patients have gradually been reported, but the findings have been inconsistent [4,5]. In addition, studies have shown that many patients with severe COVID-19 are at risk of malnutrition, and such patients have poor prognoses and outcomes [6]. A study of acutely ill patients reported that oral nutritional supplements added to a standard hospital diet initiated on the day of admission and a physical therapy program aimed at maintaining lean body mass at discharge showed no appreciable decrease in the Barthel index score (a measure of performance in activities of daily living (ADL)) at 6 months [7]. Therefore, it is important to improve physical function and activities of daily living (ADL) by using appropriate nutritional therapy in addition to rehabilitation. A concept of rehabilitation nutrition to provide high-quality care for people who develop nutritional disorders and physical dysfunction is established. The rehabilitation nutrition care process is a systematic problem-solving method that consists of five steps: (1) assessment and diagnostic reasoning; (2) diagnosis; (3) goal setting; (4) intervention; and (5) monitoring [8].

In this article, we report a case of post-acute severe COVID-19 that showed improved nutritional status and physical function using rehabilitation nutrition therapy.

## 2. Materials and Methods

### 2.1. Patient Information

A 67-year-old man was independent in ADL and worked as a builder before hospitalization. He had a history of hypertension, type 2 diabetes mellitus, and chronic kidney disease. The patient was diagnosed with COVID-19 and was admitted to another hospital 10 days before being admitted to our hospital. The patient’s condition was mild in the beginning but worsened 4 days before admission to our hospital, and oxygen therapy and antiviral drugs were administered. On day 9 after the onset of the disease, the patient was transferred to our hospital because he needed intensive care. The interventions were performed following the 1964 Declaration of Helsinki ethical standards and later amendments. The patient provided informed consent for the publication of this case report.

### 2.2. Course of Treatment

The treatment course is shown in Figure 1. On the day of admission, axial computed tomography images of the body axis showed ground-glass opacities covering a wide area of both the lungs. The patient was placed on a ventilator for 3 days after admission to our hospital. After extubation, oxygen therapy was continued, and the oxygen dosage was gradually decreased. On day 7, the patient’s antiviral medication was completed, isolation was lifted, and the patient was discharged to the general ward. On day 9, the patient’s respiratory condition worsened, and he was treated with steroid pulses for 3 days. The patient was then treated with oral steroids until day 40. On day 50, oxygen administration became unnecessary both at rest and during exertion.

### 2.3. Rehabilitation Nutrition Assessment, Diagnostic Reasoning, and Diagnosis

Sarcopenia was assessed using the Asian Working Group for Sarcopenia (AWGS) 2019 consensus (walking speed, skeletal muscle mass, and grip strength) [9]. Body composition was assessed using the bioelectrical impedance analysis using InBody 770 (InBody, Seoul, Korea). Muscle strength of the upper and lower limbs was assessed using a handgrip dynamometer and the intensive care unit (ICU) Medical Research Council (MRC) scale [10]. The modified British Medical Research Council (mMRC) dyspnea scale was used to assess dyspnea [11], and the EuroQol-5-dimensions-5-level scale was used to assess the quality of life [12]. We assessed the estimated glomerular filtration rate levels as a measure of renal function and C-reactive protein levels as a measure of inflammation. The patient’s physical and nutritional status since admission is shown in Table 1.The patient was of normal weight (body mass index (BMI) 23.8 kg/m^2^) before the onset of the disease, but at the time of ICU discharge (on day 7), the patient had lost 14% of his body weight and was severely malnourished according to the Global Leadership Initiative on Malnutrition criteria [13]. The patient was subsequently diagnosed with ICU-AW. In addition, the patient’s grip strength was 22.3 kg, and his skeletal muscle mass index (SMI) was 5.6 kg/m^2^. Applying the AWGS 2019 criteria, the patient was diagnosed with sarcopenia due to low SMI and low muscle strength on day 22.

### 2.4. Rehabilitation Nutrition Goal Setting, Intervention, and Monitoring

During the patient’s ICU stay, the main goal was to prevent ventilator-associated pneumonia and bronchial drainage by implementing remedial treatments, such as prone position therapy, which was mainly performed from day 2. After ICU discharge (day 7), we started to intervene to improve the patient’s ability to move around the bed as well as bed release time. The main goal was to reverse his sarcopenia and improve his ADL ability. Aerobic exercise was performed from day 19, and resistance training targeting the upper and lower limb muscles was performed from day 22, 40–60 min a day, 5–6 days a week. This was performed on an ergometric bicycle for 10–15 min with a target heart rate (60% load) calculated using a formula developed by Karvonen. The resistance training load was gradually increased to a target of 10 repetitions. The subjective strength during exercise was assessed using the modified Borg Scale [14], and the exercise therapy program was adjusted as necessary to ensure that general and muscle fatigue during and after exercise were less than those as per the modified Borg Scale 4.

After extubation (day 3), we confirmed that there was no problem in the patient’s swallowing function using the revised water drinking test and proceeded with oral nutrition [15]. Enteral nutrition was used for the first 5 days after admission, but sufficient energy intake was not possible due to the highly invasive nature of the disease. Even after the post-acute phase, food intake was low until day 10 due to symptoms such as respiratory distress and malaise, and the average energy intake and protein intake were 8.9 kcal/kg/day and 0.3 g/kg/day, respectively, which was unsatisfactory. Nutritional management gradually intensified from day 12, when inflammation had settled. On day 22, exercise therapy was initiated. The goal was to gain approximately 1 kg of body weight and improve the SMI over a month. From day 23, we provided one or two bottles of type A oral nutritional supplements (200 kcal, protein: 6.5 g, branched-chain amino acids: 3.5 g, Vitamin D: 1.0 g), and type B oral nutritional supplements (200 kcal, protein: 10 g, branched-chain amino acids: 1.98 g) per day, to promote muscle protein synthesis. One bottle was taken within 30 min of the end of rehabilitation (Figure 1). As a result, the daily energy intake and protein intake increased to 31.48–41.02 kcal/kg/day and 1.04–1.50 g/kg/day, respectively.

## 3. Results

Comparing the initial evaluation (on day 22) with the evaluation on discharge (day 59), the patient’s BMI had improved from 18.6 kg/m^2^ to 19.0 kg/m^2^, SMI had improved from 5.6 kg/m^2^ to 6.1 kg/m^2^, maximum grip strength had improved from 22.3 kg to 30.7 kg, gait speed had improved from 1.12 m/s to 1.20 m/s, FIM (motor items) had increased from 43 to 88 points, and the mMRC dyspnea scale score had improved from 4 to 0. In addition, there were no adverse events, such as renal dysfunction and/or glucose intolerance. The patient was discharged on day 64 and returned to work.

## 4. Discussion

In this study, the patient was diagnosed with post-acute severe COVID-19, and the results suggest that nutritional status and physical function were improved by interventions following the rehabilitation nutrition care process.

This case was a patient with post-acute severe COVID-19. Some patients with severe COVID-19 may be at a high risk of developing ICU-AW if they are admitted to an ICU and require intensive medical management, including prolonged protective lung ventilation, sedation, and the use of neuromuscular blocking agents [1]. In addition, the respiratory syndrome of COVID-19 is often associated with prolonged immobility, which leads to a loss of muscle function, resulting in sarcopenia [16]. In this case, the patient was also placed on ventilator management, sedation, and neuromuscular blocking agents, and was at high risk of developing ICU-AW. Furthermore, even after his condition stabilized, he had a long period of immobility and inadequate food intake due to symptoms such as respiratory distress, which may have resulted in his sarcopenia.

Nutritional status and physical function improved in the post-acute COVID-19 patient by interventions following the rehabilitation nutrition care process. In reviews examining the effects of rehabilitation on patients with COVID-19 [2], most studies report that COVID-19 patients are at risk of presenting with functional decline; however, evidence for the effectiveness of rehabilitation in promoting functional recovery is inconsistent and of poor quality. In contrast, it has been reported that the positive effect of exercise therapy, which was confirmed in patients with severe acute respiratory syndrome during the 2003 pandemic, may also be reproduced in COVID-19 patients, who show similar symptoms [17]. It is also reported that prevention, diagnosis, and nutritional treatment of malnutrition must be routinely included to improve the short- and long-term prognosis of COVID-19 patients [6]. As a measure of nutritional management for COVID-19 patients, energy needs to be estimated by a prediction formula based on body weight, and a protein intake of at least 1.0 g/kg (maximum 1.5 g/kg) is needed to prevent weight loss and reduce the risk of complications [16]. In addition, the mean energy intake after hospitalization is higher than that of basal energy expenditure in elderly patients with pneumonia, which affects their prognosis and ADL ability at discharge [18].

Thus, we considered it important to combine rehabilitation with appropriate nutritional therapy in post-acute severe COVID-19 patients, such as this case, who are at risk of developing long-term decline in physical function and ADL. Therefore, we intervened according to the rehabilitation nutrition care process, which is a systematic problem-solving method from the viewpoint of both rehabilitation and nutrition therapy. As a result, energy and protein intake could be met from day 22 onward, and the exercise load could be adjusted daily according to the patient’s condition, which led to improvements in factors affecting nutritional statuses, such as BMI, SMI, and physical function during hospitalization.

The main limitation of this case report is that it was validated in a single case, which seriously limits the generalizability of the results. Recovery after acute COVID-19 depends on many factors such as age and gender [19]; therefore, these factors need to be considered to further confirm the effects of rehabilitation nutrition.

## 5. Conclusions

This case showed that rehabilitation nutrition might improve the nutritional status and physical function in patients with post-acute COVID-19. It was suggested that rehabilitation nutrition might be an important treatment method for physical functional decline resulting from post-acute COVID-19, which is expected to become a global issue in the future.

## Figures and Tables

**Figure 1 healthcare-09-01034-f001:**
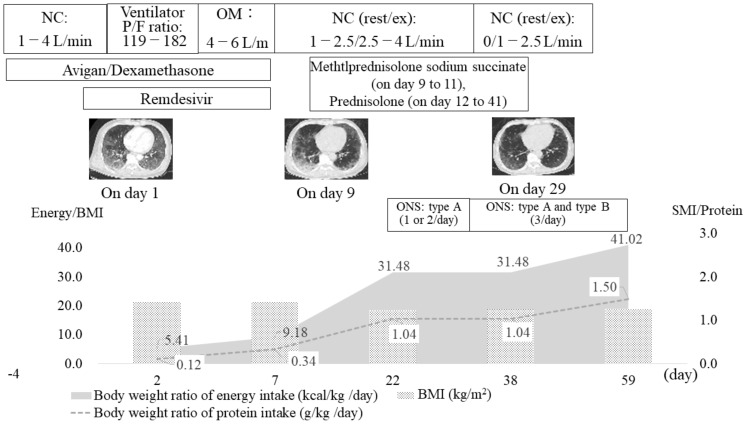
Timeline of the patient’s treatment. NC, nasal cannula; OM, oxygen mask; P/F ratio, PaO2/FiO2 ratio; BMI, body mass index; SMI, skeletal muscle mass index; ONS, oral nutritional supplements. ONS type A (energy: 200 kcal, protein: 6.5 g, branched-chain amino acids: 3.5 g, vitamin D: 1.0 g), type B (energy: 200 kcal, protein: 10 g, branched-chain amino acids: 1.98 g).

**Table 1 healthcare-09-01034-t001:** Timeline of the patient’s physical and nutritional statuses.

Number of Days of Hospitalization	2	7	22	38	59	Variation Rate
						(Day 22–59)
Body weight (kg)	61.4	60.6	53.2	53.4	54.2	1.88%
BMI (kg/m^2^)	21.5	21.2	18.6	18.7	19	2.15%
SMI (kg/m^2^)	NC	NC	5.6	5.8	6.1	8.93%
Body fat (kg)	NC	NC	12.9	12.1	12.7	−1.55%
ECW/TBW	NC	NC	0.392	0.393	0.399	1.79%
ICU MRC score (point)	NC	40	46	48	60	30.43%
Grip strength, Rt (kg)	NC	NC	21.5	NC	30.7	42.79%
Grip strength, Lt (kg)	NC	NC	22.3	NC	27.7	24.22%
Gait speed (m/s)	NC	NC	1.12	NC	1.20	6.59%
FIM, motor items (points)	13	22	43	78	88	
FIM, cognitive items (points)	12	35	12	35	35	
mMRC dyspnea scale	NC	4	4	2	0	
EQ-5D-5L	NC	NC	0.35	NC	0.842	
eGFR	69.33	61.39	74.09	83.14	89.95	
CRP (mg/dL)	6.81	1.7	0.02	1.33	0.03	

NC, not completed; BMI, body mass index; SMI, skeletal muscle mass index; ECW/TBW, extracellular water/total body water; ICU, intensive care unit; MRC, medical research council; FIM, functional independence measure; mMRC, modified British Medical Research Council; EQ-5D-5L, EuroQol-5-dimensions-5-level; CRP, C-reactive protein; eGFR, estimated glomerular filtration rate; Lt, left; Rt, right.

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
