# Peer review of "Effect of Rehabilitation Nutrition on a Post-Acute Severe COVID-19 Patient: A Case Report"

_healthcare, 2021, doi:10.3390/healthcare9081034_

Round 1

Reviewer 1 Report

Review Suggestion:

Line 40                 Delete “a” There is a concept of rehabilitation

Line 46                 using the rehabilitation nutrition therapy.

Line 51                 The patient was diagnosed with COVID-19

Line 96                 , we started to intervene with the goal of improving patient’s ability to move around the bed and bed release time (ADL).

Line 158               Thus, we considered it is important to combine

Line 224              Delete the line

Reviewer 2 Report

Dear Authors 

thank you for the opportunity to review your article "Effect of the rehabilitation nutrition on post-acute severe COVID-19 patient: A case report". Nutrition support is a very important topic and it has already been demonstrated the relevance of an adeguate intake of nutrients during ICU and intensive rehabilitation as well as in many other health conditions. The nutritional treatment, that you present as the core of the case report, consists in oral protein supplement and it didn't include supplement of vitamin D. Furthermore you missed the evaluation of dysphagia. Dysphagia is frequently underestimated and undertreated after ICU and in sarcopenia and this would have been an interesting aspect to investigate. You did not use Bioelectrical Impedence Analysis for the patient nutritional status assessment while it has a good clinical evidence. Please can you explain the reasons of these choices and discuss them? Finally in my opinion the paper is too long for a case report and table 2 is really little informative. Can you improve this aspects?

Reviewer 3 Report

The combination of physical therapy and nutritional interventions certainly helps in the recovery from the post-covid syndrome. The observations in general are acceptable. 

However, it would be of greater interest if the study has included more subjects to make it generalized as guidelines at the post-covid recovery clinics. This is particularly important considering the fact that post-covid-19 recovery highly varies between individuals and depends on a number of variable factors including immunity status, age, gender, vaccinated vs. non vaccinated, etc.,

Reviewer 4 Report

Shirado et al. have written a sound case study on the benefits of nutritional therapy on post-COVID-19 recovery. I found the paper well laid out but think it could be improved with some further background and better visualization of data which I have listed below.

More background should be given on nutritional therapy as treatment and the added benefit when used in combination of physiotherapy in COVID or other diseases (line 39).

Although figure 1 (line 60) is included in its appropriate section, it isn’t referenced in the text. For example, the lung scans could be referenced and utilized more to explain disease severity over the course of treatment.

I would suggest figure 1 (line 60) and table 1 (appx line 72) be combined into one figure for better visualization of disease response to therapy. Graphing all of the measured characteristics may not be realistic but the more heavily referenced ones like Body weight (kg). BMI, Grip Strength etc could be visualized graphically on the y axis and the points of therapy shown on the x axis.

On line 39, it would benefit the reader to have the acronym ADL expanded (explained) at its first mention, i.e. activities of daily living.
